# Digital Pathology Enables Automated and Quantitative Assessment of Inflammatory Activity in Patients with Chronic Liver Disease

**DOI:** 10.3390/biom11121808

**Published:** 2021-12-02

**Authors:** David Marti-Aguado, Matías Fernández-Patón, Clara Alfaro-Cervello, Claudia Mestre-Alagarda, Mónica Bauza, Ana Gallen-Peris, Víctor Merino, Salvador Benlloch, Judith Pérez-Rojas, Antonio Ferrández, Víctor Puglia, Marta Gimeno-Torres, Victoria Aguilera, Cristina Monton, Desamparados Escudero-García, Ángel Alberich-Bayarri, Miguel A. Serra, Luis Marti-Bonmati

**Affiliations:** 1Digestive Disease Department, Clinic University Hospital, INCLIVA Health Research Institute, 46010 Valencia, Spain; victormerim@hotmail.com (V.M.); cris_monton@hotmail.com (C.M.); M.Desamparados.Escudero@uv.es (D.E.-G.); 2Biomedical Imaging Research Group (GIBI230), La Fe Health Research Institute, 46026 Valencia, Spain; matiasgibi230@gmail.com (M.F.-P.); angel@quibim.com (Á.A.-B.); marti_lui@gva.es (L.M.-B.); 3Pathology Department, Clinic University Hospital, INCLIVA Health Research Institute, 46010 Valencia, Spain; claraalf@gmail.com (C.A.-C.); alagardac@gmail.com (C.M.-A.); Antonio.Ferrandez@uv.es (A.F.); 4Faculty of Medicine, University of Valencia, 46010 Valencia, Spain; Miguel.A.Serra@uv.es; 5Pathology Department, La Fe University and Polytechnic Hospital, 46026 Valencia, Spain; monicabauza@hotmail.com (M.B.); judithp_r@hotmail.com (J.P.-R.); 6Digestive Disease Department, Hospital Arnau de Vilanova, 46015 Valencia, Spain; anagallenperis@gmail.com (A.G.-P.); salvaben@hotmail.com (S.B.); 7CIBERehd, Centro de Investigación Biomédica en Red en Enfermedades Hepáticas y Digestivas, Instituto de Salud Carlos III, 28029 Madrid, Spain; toyagui@hotmail.com; 8Pathology Department, Hospital Arnau de Vilanova, 46015 Valencia, Spain; victorpuglia@gmail.com; 9Hepatology and Liver Transplantation Unit, La Fe University and Polytechnic Hospital, 46026 Valencia, Spain; mgimeno33@gmail.com; 10Quantitative Imaging Biomarkers in Medicine, QUIBIM SL, 46021 Valencia, Spain; 11Radiology Department, La Fe University and Polytechnic Hospital, 46026 Valencia, Spain

**Keywords:** digital pathology, inflammation, nonalcoholic fatty liver disease, chronic hepatitis

## Abstract

Traditional histological evaluation for grading liver disease severity is based on subjective and semi-quantitative scores. We examined the relationship between digital pathology analysis and corresponding scoring systems for the assessment of hepatic necroinflammatory activity. A prospective, multicenter study including 156 patients with chronic liver disease (74% nonalcoholic fatty liver disease-NAFLD, 26% chronic hepatitis-CH etiologies) was performed. Inflammation was graded according to the Nonalcoholic Steatohepatitis (NASH) Clinical Research Network system and METAVIR score. Whole-slide digital image analysis based on quantitative (I-score: inflammation ratio) and morphometric (C-score: proportionate area of staining intensities clusters) measurements were independently performed. Our data show that I-scores and C-scores increase with inflammation grades (*p* < 0.001). High correlation was seen for CH (*ρ* = 0.85–0.88), but only moderate for NAFLD (*ρ* = 0.5–0.53). I-score (*p* = 0.008) and C-score (*p* = 0.002) were higher for CH than NAFLD. Our MATLAB algorithm performed better than QuPath software for the diagnosis of low-moderate inflammation (*p* < 0.05). C-score AUC for classifying NASH was 0.75 (95%CI, 0.65–0.84) and for moderate/severe CH was 0.99 (95%CI, 0.97–1.00). Digital pathology measurements increased with fibrosis stages (*p* < 0.001). In conclusion, quantitative and morphometric metrics of inflammatory burden obtained by digital pathology correlate well with pathologists’ scores, showing a higher accuracy for the evaluation of CH than NAFLD.

## 1. Introduction

Chronic liver disease is a major cause of morbidity, mortality, and health care resource utilization worldwide [1]. Liver biopsy remains the reference method to evaluate the extent and distribution of the histological features that define disease severity. However, traditional histological evaluation is based on subjective and semi-quantitative scoring systems that are prone to interobserver variability [2]. Additionally, the use of grading and staging systems creates discrete categories from pathological processes that are a histological continuum [3]. 

Histologically identified necroinflammatory activity has prognostic significance in the management of chronic liver diseases [4,5,6]. Systems used to grade the intensity of injury in chronic hepatitis include morphological assessment of both inflammation and hepatocellular necrosis. Generally, the inter-reader reliability agreement between pathologists is lower for necroinflammatory changes than for other histological features [2,7]. In practice, pathologists must decide which of several grading systems to use, according to the clinical context and histological findings. Scoring systems are disease-specific and a system suitable for a particular disease process is unlikely to be suitable for a different process. For example, the lobular pattern of inflammation seen in steatohepatitis is different from the portal distribution seen in chronic viral or autoimmune hepatitis, and therefore different grading systems are required. 

To overcome these limitations, emerging computer-assisted digital image analysis technologies have been used to assess histological features in an objective, reliable, and quantitative way [8]. Digital pathology is the scanning of tissue slides to create digital images, allowing quantitative and automated analysis [9]. This technique plays a fundamental role in cancer diagnosis and treatment assessment [10]. Adoption of computerized analysis based on machine learning algorithms has shown reproducible results for grading histologic features in liver disease [11,12]. Such methods are not presently in clinical use because they require high-resolution images and specialized equipment [13]. With the increasing availability of digital images, simple and user-friendly image processing software is rather required for digital pathology analysis. In this regard, previous studies have used different programs (Zeiss Axiovision or MATLAB) to quantify the proportionate area of collagen (based on stained colour thresholds) and correlated with fibrosis stages [14,15,16]. However, assessment of other important histological features, such as necroinflammatory activity, using a morphological analysis approach is still lacking. 

We hypothesized that automated digital image analysis can accurately quantify necroinflammatory activity in biopsies from participants with chronic liver diseases. The primary aim of this study was to identify and quantify hepatic inflammatory areas based on immunohistochemical and morphometric features and to evaluate its correlation with the corresponding pathologists´ grading scores in a well characterized cohort of patients with chronic liver disease.

## 2. Materials and Methods

### 2.1. Study Design and Subjects

A cross-sectional, multicenter, prospective study was performed between 2017 and 2021. This study recruited consecutive chronic liver disease patients who had undergone clinically indicated liver biopsy at the Hepatology Units of three participating Spanish centers. Target chronic liver diseases were NAFLD, AIH, and viral hepatitis. Criteria for inclusion were age over 18 years and liver biopsies greater than 15 mm in length and containing at least six portal tracts. Sample biopsies containing tumoral cells and histological diagnosis of competing liver diseases including alcoholic hepatitis and primary biliary cholangitis (PBC) were excluded. Baseline characteristics, including clinical and laboratory parameters, were collected. We initially recruited 162 participants with chronic liver disease and biopsy. Indications for liver biopsy were to determine the degree of fibrosis or inflammatory activity (40%), uncertain or mixed etiology (38%), confirm HAI diagnosis (12%), and inclusion in clinical trials (10%). After excluding inadequate histological samples (*n* = 3) and cases with histological features of PBC non-overlapped with AIH (*n* = 3), the final series included 156 patients. Among them, 74% (*n* = 116) had NAFLD and 26% (*n* = 40) had CH. Informed consent was provided by all subjects prior to their participation in the study. The study protocol conformed to the ethical guidelines of the 1975 Declaration of Helsinki and received ethical approval by the Institution Review Board of the three participating hospitals (2016/209; 2017/0031/PI, and 29/2019). A portion of the included participant data (*n* = 78) has already been published regarding technical developments of digital pathology [17]. The prior study used a simpler methodology for digital image analysis and did not evaluate other chronic hepatitis etiologies beyond NAFLD.

### 2.2. Histological Analysis

Percutaneous biopsies were obtained with a semi-automatic 16G, two-step needle. After formalin fixation (10% buffered), paraffin-embedded tissue sections (4 µm-thick) were stained with hematoxylin and eosin (H&E). Immunohistochemical staining was performed with an automated system Autostainer Dako Omnis (Agilent Technologies, DAKO) on paraffin-embedded tissue sections of the same block. Prediluted monoclonal mouse antibody against CD45 (Leucocyte Common Antigen, Clone 2B11 + PD7/26, DAKO) was used to highlight inflammatory activity for subsequent digital analysis. CD45 stains lymphocytes, basophils, eosinophils, neutrophils, macrophages, monocytes, mast cells, plasma cells and dendritic cells [18]. Conventional H&E-stained sections were not used for digital analysis because of possible methodological errors caused by the identification of non-inflammatory cells spaces, such as hepatocytes and cholangiocytes nucleus (details in Section A.1). Immunohistochemistry for CD45 was performed at the time of biopsy reception and evaluation for clinical diagnosis. Inflammation grades throughout this study refer to inflammatory activity grades and lobular inflammation grades as scored by the METAVIR and the NASH Clinical Research Network (NASH-CRN) scoring systems, respectively [19,20]. For patients with nonalcoholic fatty liver disease (NAFLD), the NASH-CRN scoring system was used to grade lobular inflammation from 0 to 3 and fibrosis from F0 to F4 [19]. For patients with CH etiologies (HBV, HCV, or autoimmune hepatitis), the METAVIR scoring system was used to grade inflammatory activity from 0 to 3 and fibrosis stages from F0 to F4 [20]. Steatosis grading for all cases was performed on an ordinal scale from S0 to S3 [19]. Patients with NAFLD were categorized as either simple steatosis or NASH, defining NASH as by the presence of hepatic steatosis with lobular inflammation and definite ballooning with or without fibrosis. Patients with CH were also categorized in two disease categories according to their METAVIR activity score: none/low (0–1) vs. moderate/severe (2–3) inflammation activity. Disease categories were selected based on the risk of NASH and moderate necroinflammatory activity progression to advanced fibrosis [21,22]. For each biopsy, the total length and number of portal tracts were recorded. Histological assessment was performed centrally by two experienced pathologists (C.A.C. and A.F., each with >10 years of experience in the field of liver pathology), blinded to clinical data and digital image analysis results.

### 2.3. Digital Image Analysis

Whole-slide samples were scanned (Ventana iScan HT slide scanner, Roche, Ventana Medical Systems, Inc., Oro Valley, AZ, USA) using a 40× magnification objective and a calibrated camera (4000 × 4000 pixels = 1 mm^2^). The images were stored in 24-bit RGB colour format. Digital image analysis was performed at the highest resolution level using an in-house automatic tool developed using MATLAB software (MathWorks, version R2016a) [17]. The analysis included a sequential workflow of six-steps: (a) detection of the tissue area and removal of the background; (b) colour standardization and transformation to *CIELab* format to reduce variability between images [23]; (c) segmentation based on colour threshold to detect CD45+ stained cells; (d) mask dilatation of 4-pixels to draw together neighboring inflammatory cells forming foci (morphologic closing); (e) removal of segmented areas <0.0032 mm^2^ to exclusively detect clusters of inflammatory cells; and (f) removal of unstained CD45+ areas from the final extracted mask to avoid overestimation. The optimal colour threshold for the detection of stained cells was established with the consensus of pathologists, taking one case as reference. The rest of the cases were normalized to this case using the Reinhard normalization [23]. Hence, all the cases presented a similar distribution of colour intensities. Mask dilatation value was selected after a series of trial and improvement analysis involving pathologists, hepatologists, and biomedical engineers (details in Section A.2). Distribution of the segmented area (lobular and/or portal inflammation) was not considered. The whole process is illustrated in Figure 1. This computational algorithm based on immunohistochemical colour and morphological features identified CD45+ staining intensity areas and clusters of staining intensities, allowing the quantification of the following scores: (1)$$\mathrm{Inflammation}\;\mathrm{proportionate}\;\mathrm{area}\;\left(I-\mathrm{score}\right)=\frac{{\mathrm{segmented}\;\mathrm{area}\;}_{\mathrm{total}\;\mathrm{tissue}}}{{\mathrm{area}\;}_{\mathrm{total}\;\mathrm{tissue}}\;}\times \;100\;\left(\mathrm{in}\;\%\right)$$
(2)$$\mathrm{Clusters}\;\mathrm{proportion}\;\left(C-\mathrm{score}\right)=\frac{{\mathrm{number}\;\mathrm{of}\;\mathrm{segmented}\;\mathrm{foci}\;}_{\mathrm{total}\;\mathrm{tissue}}}{{\mathrm{area}\;}_{\mathrm{total}\;\mathrm{tissue}}}$$

The analyses were repeated with the open-source digital image analysis software QuPath v0.3 (https://qupath.github.io/ (accessed on 16 November 2021) [24]. The CD45 DAB-stained images were imported into QuPath and the total tissue area was detected using the wand detection tool. The uniform intensity staining pattern of CD45 lends itself to assessment by a simple density method within QuPath (DAB threshold). To compare our algorithm results with an alternative widely used software, we calculated CD45 positive area divided by the total analyzed tissue area, expressed as a percentage (QuPath’s positive staining detection algorithm).

### 2.4. Statistical Analysis

Data were analyzed using SPSS V25.0 software package (IBM) and MedCalc Statistical Software 19.4 (MedCalc Software, Ostend, Belgium), by one author (D.M.A., with >5 years of experience in statistics). Categorical data were expressed as frequencies (%), and quantitative data as the mean and SD or median and interquartile range (IQR). Comparisons between digital image analysis quantitative data and histological groups were assessed using ANOVA or Kruskal-Wallis test followed by multiple comparisons between groups using post hoc Bonferroni or Tukey’s range test, as required. The Spearman test (*ρ*) was used to calculate correlations between inflammation scores and digital pathology. Comparisons between digital pathology measurements and disease categories were assessed using a T-student or Mann-Whitney U test, as required. Comparisons between paired distributions of NAFLD and CH etiology for digital pathology data were assessed with the Wilcoxon matched pairs test stratified by histological grade. The receiver operating characteristic (ROC) curves and areas under the ROC curve (AUC) were calculated for each set of dichotomized groups of different histological grades of inflammation and the optimal threshold for diagnosis of NASH and moderate/severe activity (inflammation grades 2–3) were established. For each ROC analysis, the optimal cut-off value was selected using Youden’s index to maximize sensitivity and specificity. The DeLong test was used to compare the AUCs of software’s performance for specific grades of lobular inflammation and METAVIR score. A logistic regression model was constructed to examine factors associated with NASH. Digital pathology data and liver function tests exhibiting *p*-value < 0.10 in the univariate model were incorporated into the final stepwise logistic regression analysis. Linear regression analysis was performed to determine the correlation (Pearson correlation coefficient [*r*]) between MATLAB and QuPath data. All tests were two-sided and a *p*-value < 0.05 was considered statistically significant.

## 3. Results

### 3.1. Patients and Histological Characteristics

The clinical and biochemical characteristics of the cohort are summarized in Table 1. In brief, 58% were female (*n* = 91) with a mean age of 55 ± 12 years and most were overweight (mean BMI of 28.7 kg/m^2^, 43% (*n* = 68) obese). The histopathological features of participants are summarized in Table 2. Median biopsy length was 20 (17–23) mm and included 10 (8–13) portal tracts. Histological review showed that 23% had mild steatosis (S1, *n* = 36), 22% had moderate steatosis (S2, *n* = 34), and 22% presented severe steatosis (S3, *n* = 35). In NAFLD cases, 57% (*n* = 67) had NASH, and in patients with chronic hepatitis, 37% had significant fibrosis (≥F2, *n* = 15).

### 3.2. Digital Image Analysis and Inflammation Scoring Systems

Distribution of digital pathology values across histological grades is shown in Table 3. I-score and C-score increased with inflammation grades (*p* < 0.001). There were statistically significant differences between overall I-score (*p* = 0.008) and C-score (*p* = 0.002) quantified for NAFLD and CH cases. Analyzing individual grades separately, higher measured values were obtained among CH tissue sections for moderate and severe inflammation (mean I-score increase of 3.5% and 3.2%, mean C-score increase of 11.8 and 17.3 foci/mm^2^, respectively). Within NAFLD cases, a moderate correlation was seen between digital pathology and histological lobular inflammation (*ρ* = 0.50 for I-score, *ρ* = 0.53 for C-score; *p* < 0.001), discrimination only being possible between lobular inflammation grade 0 vs. grades 1–3, and grades 0–1 vs. grade 3 (Figure 2). A higher correlation was seen for CH METAVIR histologic score (*ρ* = 0.85 for I-score, *ρ* = 0.88 for C-score; *p* < 0.001), differentiating between grades 0–1 vs. grades 2–3 (Figure 2). Although QuPath metrics also increased with grading systems (*p* < 0.001) and a strong correlation was seen with I-score (r = 0.78; *p* < 0.001) and C-score (r = 0.76; *p* < 0.001), the accuracy of our MATLAB software parameters was higher (details in Section A.3). The AUC of I-score for detecting moderate/severe inflammation (≥grade 2) was 0.72 [95%CI, 0.62–0.82] for NAFLD and 0.97 (95%CI, 0.92–1.00) for CH (Table 4). The best I-score cut-off values for detecting moderate/severe inflammation was 0.6% (86% sensitivity, 70% specificity), and 2.0% (94% sensitivity and 91% specificity) for NAFLD and CH, respectively. The DeLong test showed no difference when comparing the AUCs of I-score and C-score for detecting different inflammatory grades. 

### 3.3. Relationship between Different Histological Features

Distribution of digital pathology values across NAFLD and chronic hepatitis disease categories are summarized in Table 5. Patients with NASH and moderate/severe activity in chronic hepatitis had a higher proportion of C-score compared to patients with simple steatosis and none/low chronic hepatitis activity, respectively. In the multivariable analysis adjusted by liver function tests, I-score and C-score were both independent factors for predicting NASH (Table 6). Using C-score values, the AUC for detecting NASH was 0.75 (95%CI, 0.65–0.84) and for moderate/severe CH activity diagnosis it was 0.99 (95%CI, 0.97–1.00) (Figure 3). The optimal cut-off value for detecting NASH was 4.5 foci/mm^2^, with 79% sensitivity and 65% specificity, whereas for moderate/severe CH activity it was 9.6 foci/mm^2^, with 94% sensitivity and 96% specificity. Higher overall measurements were obtained for individual with moderate/severe CH activity than NASH (mean I-score increase of 4.4% and mean C-score increase of 17.2 foci/mm^2^; *p* < 0.001). No such differences were seen within disease categories of simple steatosis and none/low CH activity (Table 5).

Digital pathology CD45+ measurements also increased significantly with fibrosis stages (*p* < 0.001). Specifically, a linear association was seen between fibrosis stages and both I-score and C-score, stratified by liver disease etiology (Figure 4). For the NAFLD cohort, I-score (0.6 ± 0.5, 1.0 ± 0.7, 1.5 ± 1.6, 2.1 ± 1.4, and 2.6 ± 1.3, F0–F4, respectively) was able to differentiate between F0 vs. F2–4, F1 vs. F3–4, F2 vs. F4 (*p* < 0.001); and C-score (2.6 ± 1.9, 4.6 ± 2.6, 5.9 ± 2.5, 9.2 ± 3.1, and 11.7 ± 4.6, F0–F4, respectively) was able to differentiate between F0 vs. F2–4 and F1–2 vs. F3–4 (*p* < 0.001). For CH cohort, I-score (0.5 ± 0.5, 3.4 ± 2.8, 5.1 ± 2.9, 7.1 ± 3.1, and 7.0 ± 0.6, F0–F4, respectively) was able to differentiate between F0 vs. F1–4, F1 vs. F3–4, F2 vs. F4 (*p* < 0.001); and C-score (3.1 ± 3.5, 9.0 ± 4.6, 18.6 ± 12.9, 30.5 ± 16.7, and 48.8 ± 17.9, F0–F4, respectively) was able to differentiate between F0 vs. F2–4, F1 vs. F3–4, F2 vs. F4 (*p* < 0.001).

## 4. Discussion

There is a clear need for objective quantification of histological features in liver biopsies to circumvent classical microscopy techniques limitations. In this work, we investigated the use of digital pathology and image processing for the quantitative and morphometric assessment of necroinflammatory activity features in chronic liver diseases. Importantly, we found that digital image analysis-derived parameters are able to accurately differentiate between grades of inflammation. Compared to NAFLD, chronic hepatitis etiologies had higher inflammatory burden as assessed by staining intensity areas, and digital pathology-derived metrics showed stronger correlation with the METAVIR scoring system.

Digital pathology imaging analysis is emerging as a promising tool and has already been demonstrated to be a reliable and precise technique for quantitative histological assessment [8]. Its main clinical application has been the quantification of collagen fibers amount for continuous staging of hepatic fibrosis and prediction of clinical outcomes in chronic liver diseases, including hepatitis C and alcoholic liver disease [14,25,26,27]. With recent changing trends in chronic liver disease etiologies, NAFLD biopsies have also been evaluated with digital image analysis for fibrosis and steatosis detection and quantification [15,16,17]. However, to date the quantitative evaluation of necroinflammatory histologic activity and the potential role of morphometric analysis based on qualitative patterns of injury such as cell closing in clusters has been limited. 

To grade inflammatory activity, simple and clinically validated scoring systems based on the underlying disease were used in the current study [19,20]. Lobular inflammation grade is based on the number of necroinflammatory foci in a given area with clusters of mainly lymphocytes and monocytes [28]. Periportal interface hepatitis is usually recognized as an extension of lymphoplasmacytic inflammation foci into the surrounding parenchyma to replace hepatocytes lost by apoptosis [7]. Because the lesions can overlap and vary considerably from one zone to the next, the utility of a single, disease-specific grading scheme is limited, and reproducibility between pathologists can be hampered. To overcome these limitations, digital pathology offers a broad approach regardless of liver disease damage (patterns of injury, localization, and etiology). For digital image analysis, this study used immunohistochemistry CD45, one of the most abundant leucocyte surface glycoproteins of haematopoietic lineage cells [18]. Colour is considered a source of histopathological image variation with the widely used H&E staining, but the effects of pre-processing methods are not entirely clear yet [29]. This is the reason why we used CD45, an IHC that lends itself towards a more uniform intensity staining pattern. Color normalization was applied as previously described in publications of artificial intelligence-assisted technology to assess liver biopsies [30]. 

The digital pathology software used was previously developed in NAFLD patients [17]. The present study was performed in a well-characterized cohort of 156 patients with a wide range of chronic liver diseases including autoimmune and viral hepatitis. Current image analysis is more advanced than prior colour-space algorithm given that a sequential mask dilatation and size thresholding is now applied (Figure 1). This workflow allows the quantification of inflammatory burden with the developed I-score and C-score. This CD45+ staining intensity area analysis has lower values than alternative software that only use colour-space thresholding: average of 1.1 (0.6–2.3) with current algorithm vs. 5.4 (4.2–7.1) with our prior algorithm and 7.2 (4.1–10.2) with QuPath. The new scores classify better the disease activity, reflecting the importance of morphometric assessment. 

Higher I-scores and C-scores were seen for CH etiologies compared to NAFLD. Different values according to liver disease etiologies have also been reported for collagen proportionate area [31]. This can be explained because inflammatory diseases like viral and autoimmune hepatitis have a faster rate of progression compared to NAFLD [32]. The present study provides evidence that digital pathology-derived measurements of inflammation staining areas (total area ratio and morphologic closing in clusters) increases with inflammation grades. Correlation of lobular inflammation grades with these CD45+ staining intensity metrics was slightly higher in our study (*ρ* = 0.50–0.53) compared to previous publications that used supervised machine learning techniques (*ρ* = 0.35–0.45) [11,12,13]. This result reinforces the utility of simple computational algorithms based on imaging processing and such simplicity may facilitate its clinical adoption. Our MATLAB software also showed better performance than QuPath positive staining detection algorithm (see Section A.3). The counting different cell types command should be explored in the future to evaluate whether it can add accuracy for disease activity classification. 

It is important to diagnosis individuals with NASH for inclusion in clinical trials given its prediction of adverse outcomes [5]. NASH is a complex process that involves a number of disturbances including oxidative stress, increased hepatocellular apoptosis, and inflammation. Prior investigations focusing on identifying serum biomarkers for detecting NASH consistently obtained poor results [33]. Recently, a combined biomarker of AST and liver stiffness measurement by transient elastography (FAST score) has shown good accuracy for identifying patients at risk of progressive NASH [34]. In our study, AST was only associated with NASH in the univariate analysis. In the multivariable analysis, none of the liver function tests, only digital pathology metrics, were independent factors for predicting NASH (Table 6). The C-score performed better than I-score to classify NASH, adding valuable information to previous results that just analyzed inflammatory proportionate areas [17]. This score based on clusters of stained intensity areas could help to diagnose a NASH given the low inter-observer agreement between pathologists and that non-NASH cases still have some inflammatory degree in the liver [2,35,36]. The C-score represents a morphological metric associated with inflammatory changes and key features related to NAFLD: clusters of different cells including lymphocytes, plasma cells, monocytes, polymorphonuclears, eosinophils, and Kupffer cells, all of them positive for CD45 staining [28]. 

Diagnostic performance of both I-score and C-score was better in cases with a CH etiology. This can be attributed to the less prominent degree of inflammation among NAFLD cases (Table 3). Activity classification of moderate/severe CH inflammation was well detected with digital image analysis metrics (AUC= 0.97–0.99). Activity CH disease categories are better distinguished than the NAFLD spectrum perhaps because NASH represents a complex pathological entity where other histological features can play a role [28]. Finally, I-score and C-score also increased with fibrosis stage (Figure 4). In close agreement with other recent studies, these data support the dynamic of inflammatory cells infiltration in the liver across disease progression of fibrosis to cirrhosis [36,37]. 

Our study has some limitations that must be considered for future research perspectives. First, CD45 non-specifically stains a range of different haematopoietic cells. Other more specific markers, such as CD68 and CD163 for macrophage/Kupffer cells, could be addressed in future studies given its role in NAFLD severity [38]. Moreover, our study did not differentiate between lobular and portal inflammation. This can be important given that portal chronic inflammation is related to NAFLD severity [39]. Cells detection and type classification is possible with machine learning algorithms and certain software [11,12,13,24]. Our digital pathology analysis is based on image processing and did not used advanced deep learning methodology. Future studies could quantify cellular infiltrates as positively stained cells/negatively stained cells, which could add accuracy to disease classification. The cross-sectional nature does not allow the assessment of changes in the proportions of inflammation burden over the time course of the disease. Future studies could correlate digital image analysis measurements with clinical outcomes. Traditional grading systems may be insufficiently granular to detect small clinically relevant changes in inflammation that could be detected by digital pathology methods [40]. In the future, digital pathology analysis may offer an objective evaluation of therapeutic drugs effects and play a key role in clinical trials. 

## 5. Conclusions

Digital pathology with computational analysis allows an automated, quantitative, and morphometric assessment of hepatic necroinflammatory activity. Our methodology shows good correlation with scoring by pathologists and reveals a better performance among chronic hepatitis etiologies than NAFLD. Developed I-scores and C-scores based on staining intensity areas offer promise for further development of automatic quantification as a potential aid to pathologists evaluating chronic liver disease biopsies in clinical practice and clinical trials.

## Figures and Tables

**Figure 1 biomolecules-11-01808-f001:**
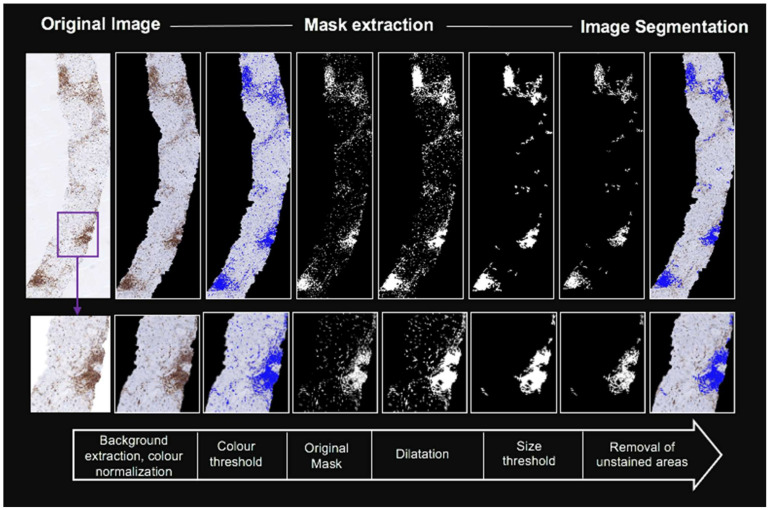
Digital image analysis workflow in a biopsy specimen with NASH diagnosis. Mask extraction was based on colour detection, size threshold, and morphological analysis. Bottom row shows a small region of interest to display details of the image processing algorithm, although the analysis was performed on the whole slice. Image segmentation of the represented case determined inflammation proportional area (I-score: 4.9% CD45+) and clusters proportion (C-score: 12.2 foci/mm^2^).

**Figure 2 biomolecules-11-01808-f002:**
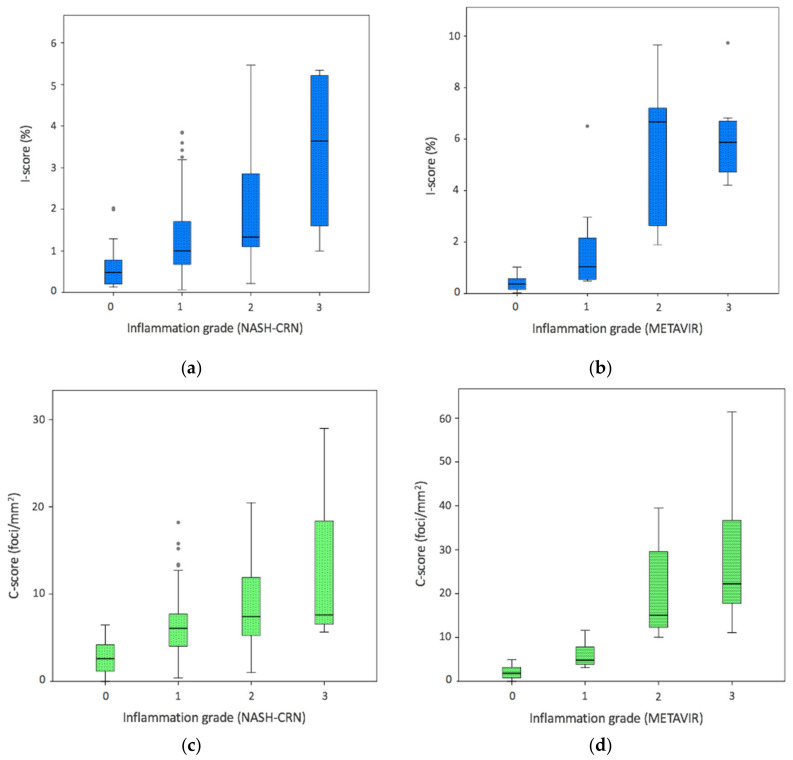
Distribution of digital pathology data across necroinflammatory activity histological scores. Boxplot of (**a**) I-score in NAFLD cohort, (**b**) I-score in chronic hepatitis cases, (**c**) C-score in NAFLD, and (**d**) C-score in chronic hepatitis. Blue boxes correspond to I-score (%CD45+) and green boxes to C-score (clusters proportion) data.

**Figure 3 biomolecules-11-01808-f003:**
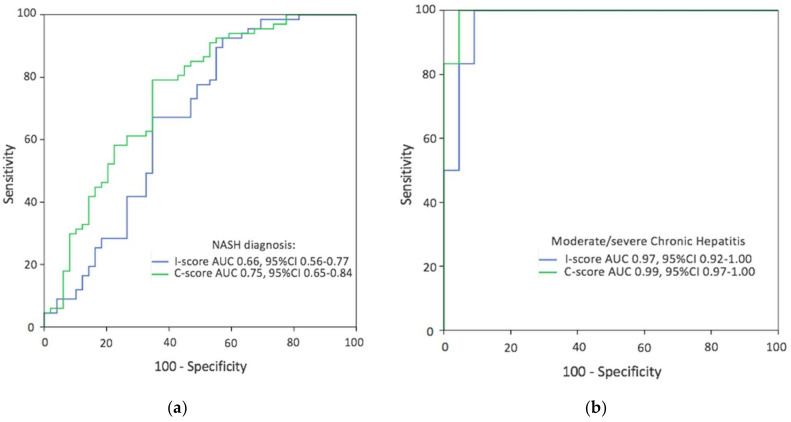
Receiver operating characteristic curves of digital pathology data for discriminating the presence of (**a**) NASH in NAFLD population, and (**b**) moderate/severe inflammation in chronic hepatitis cases. The blue line corresponds to I-score (%CD45+), and the green line corresponds to C-score (clusters proportion).

**Figure 4 biomolecules-11-01808-f004:**
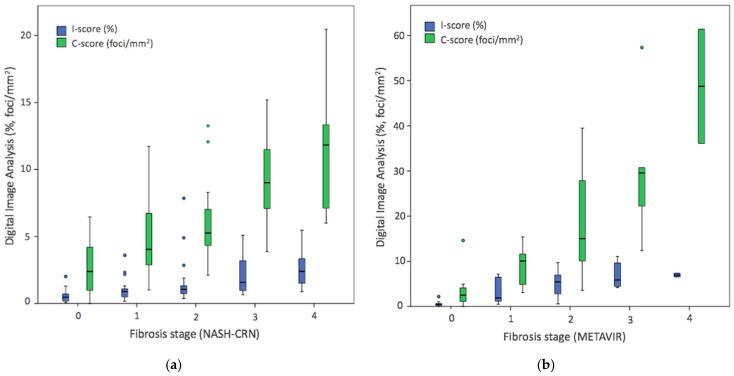
Distribution of digital pathology data across fibrosis histological scores. Boxplot of (**a**) I-score and C-score in NAFLD cohort, (**b**) I-score and C-score in chronic hepatitis cases. Blue boxes correspond to I-score (%CD45+) and green boxes to C-score (clusters proportion) data.

**Table 1 biomolecules-11-01808-t001:** Characteristics of the included population.

Characteristic	Overall Cohort (*n* = 156)
Female sex	91 (58%)
BMI (kg/m^2^) Hypertension Diabetes Dyslipidemia	28.7 ± 5.1 71 (45%) 54 (34%) 99 (63%)
Liver disease etiology	
• NAFLD	116 (74%)
• AIH	32 (21%)
• Viral	8 (5%)
Platelet count (×10^9^/L)	231 ± 76
AST (U/L)	39 (29–59)
ALT (U/L) GGT (U/L)	49 (36–75) 81 (50–174)
Total bilirubin (mg/dL) Albumin (g/dL) INR	0.6 (0.4–0.8) 4.4 (4.2–4.6) 1.0 (1.0–1.1)
Triglycerides (mg/dL)	116 (80–166)
Total cholesterol (mg/dL)	189 ± 41

Note: Data reported as mean ± standard deviation when normal distribution, median (IQR) when skewed distribution or count (percentage). Abbreviations: AIH, autoimmune hepatitis; ALT, alanine aminotransferase; AST, aspartate aminotransferase; BMI, body mass index; INR, international normalized ratio; NAFLD, nonalcoholic fatty liver disease.

**Table 2 biomolecules-11-01808-t002:** Histopathological characteristic distribution of the study population.

Histology		NAFLD (*n* = 116)	CH (*n* = 40)
Feature	Grade/Stage	NASH-CRN Score	METAVIR Score
Inflammation	Grade 0	26 (22%)	14 (34%)
	Grade 1	58 (50%)	8 (20%)
	Grade 2	28 (24%)	9 (23%)
	Grade 3	4 (3%)	9 (23%)
Fibrosis	Stage 0	26 (22%)	16 (40%)
	Stage 1	28 (24%)	9 (23%)
	Stage 2	26 (22%)	8 (20%)
	Stage 3	18 (16%)	5 (12%)
	Stage 4	18 (16%)	2 (5%)

Note: Data reported as count (percentage). Abbreviations: CH, chronic hepatitis; NAFLD, nonalcoholic fatty liver disease; NASH-CRN, nonalcoholic steatohepatitis clinical research network.

**Table 3 biomolecules-11-01808-t003:** Summary of I-score (%CD45+) and C-score (clusters proportion) results (mean ± SD) for each grade of inflammation. The *p*-value indicates the significance of paired distributions comparison between NAFLD and chronic hepatitis.

Etiology	Histology Grade	I-Score (%)	*p*-Value	C-Score (foci/mm^2^)	*p*-Value
NAFLD	NASH-CRN 0 METAVIR 0	0.6 ± 0.5	0.23	2.7 ± 1.8	0.38
CH		0.4 ± 0.3		2.2 ± 1.6	
NAFLD	NASH-CRN 1 METAVIR 1	1.5 ± 1.2	0.58	6.8 ± 3.8	0.56
CH		1.8 ± 2.1		5.9 ± 3.0	
NAFLD	NASH-CRN 2 METAVIR 2	1.9 ± 1.4	0.005	8.5 ± 4.8	0.015
CH		5.4 ± 2.8		20.3 ± 11.5	
NAFLD	NASH-CRN 3 METAVIR 3	3.4 ± 2.1	0.04	12.5 ± 11.1	0.06
CH		6.6 ± 2.4		29.8 ± 18.6	

Abbreviations: CH, chronic hepatitis; NAFLD, nonalcoholic fatty liver disease; NASH-CRN, nonalcoholic steatohepatitis clinical research network.

**Table 4 biomolecules-11-01808-t004:** Diagnostic accuracy of digital image analysis: raw performance parameters of I-score (%CD45+) and C-score (clusters proportion) for different etiologies of chronic liver disease stratified by histological grade. The DeLong *p*-value compares the AUCs of the software performance for specific grades of NASH-CRN and METAVIR scores.

Etiology	Grades	I-Score Cut-Off (%)	I-Score AUC (95%CI)	C-Score Cut-Off (foci/mm^2^)	C-Score AUC (95%CI)	DeLong *p*-Value
NAFLD	NASH-CRN ≥ 1 METAVIR ≥ 1	>0.6	0.83 (0.74–0.92)	>4.5	0.87 (0.80–0.94)	0.20
CH		>0.9	0.96 (0.92–1.00)	>4.4	0.96 (0.92–1.00)	0.99
NAFLD	NASH-CRN ≥ 2 METAVIR ≥ 2	>0.9	0.72 (0.62–0.82)	>6.6	0.72 (0.61–0.82)	0.94
CH		>2.0	0.97 (0.92–1.00)	>9.6	0.99 (0.97–1.00)	0.22
NAFLD	NASH-CRN ≥ 3 METAVIR ≥ 3	>2.2	0.83 (0.64–0.99)	>7.4	0.75 (0.56–0.93)	0.16
CH		>4.4	0.86 (0.74–0.97)	>12.3	0.91 (0.82–0.99)	0.27

Abbreviations: AUC, area under the curve; CH, chronic hepatitis; NAFLD, nonalcoholic fatty liver disease, NASH-CRN, nonalcoholic steatohepatitis clinical research network.

**Table 5 biomolecules-11-01808-t005:** Distribution of digital pathology data across disease categories of NAFLD (simple steatosis and NASH) and chronic hepatitis (none/low and moderate/severe inflammation).

Digital Pathology	NAFLD (*n* = 116)			Chronic Hepatitis (*n* = 40)		
Measurements	Simple Steatosis	NASH	*p*-Value	None/Low	Moderate/ Severe	*p*-Value
I-score (%)	1.2 ± 1.3	1.6 ± 1.3	0.08	0.9 ± 1.4	6.0 ± 2.6	<0.001
C-score (foci/mm^2^)	4.6 ± 3.9	7.9 ± 4.8	<0.001	3.6 ± 2.8	25.1 ± 15.8	<0.001

Abbreviations: NAFLD, nonalcoholic fatty liver disease; NASH: nonalcoholic steatohepatitis.

**Table 6 biomolecules-11-01808-t006:** Predictive factors for NASH, as assessed using univariate and multivariate regression models.

	Univariate Analysis		Multivariate Analysis	
Parameter	OR (95%CI)	*p*-Value	OR (95%CI)	*p*-Value
AST ^1^	1.02 (1.0–1.04)	0.025	1.01 (0.99–1.03)	0.28
ALT	1.01 (0.99–1.02)	0.37		
GGT	1.0 (0.99–1.0)	0.26		
Total bilirubin	0.48 (0.17–1.39)	0.17		
Albumin	0.86 (0.33–2.23)	0.74		
INR	1.01 (0.97–1.04)	0.29		
Platelets	1.0 (0.99–1.01)	0.75		
I-score ^2^	1.32 (0.97–1.80)	0.08	1.71 (1.04–2.81)	0.03
C-score ^2^	1.24 (1.09–1.39)	<0.001	1.41 (1.16–1.70)	0.01

Abbreviations: ALT, alanine aminotransferase; AST, aspartate aminotransferase; INR, international normalized ratio, I-score, C-score. ^1^ Significant factor, as assessed with univariate analysis. ^2^ Significant factors, according to both univariate and multivariate analyses.

## Data Availability

The data and the MATLAB software code that support the findings of this study will be available from the corresponding author upon reasonable request.

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
