# Peer review of "Digital Pathology Enables Automated and Quantitative Assessment of Inflammatory Activity in Patients with Chronic Liver Disease"

_biomolecules, 2021, doi:10.3390/biom11121808_

Round 1
Reviewer 1 Report
The authors describe the use of simple colorspace thresholding of images of CD45 immunostaining of medical liver biopsies to quantify the inflammatory burden. Digital pathology-derived metrics are then assessed for their abilities to classify disease activity defined by traditional alternative subjective assessment.
To what extent this work represents new data compared with that published by the authors previously – citation [16] in APT – is not clear. Certainly it appears, without further clarification, that the NAFLD cohort in this study may be the same or significantly overlap with that previously published, and the CD45 quantification of inflammatory area having also been published before. The cluster counting is new in this paper but doesn’t add much above inflammatory proportion and the subjective metric for comparison in this paper is derived from NAS rather than SAF scoring. The only real reference to the prior work in Discussion – ‘This data adds valuable information to previous results that only appreciated differences between inflammation grades when large influx of inflammatory cells infiltrates the liver [16]’ – is vague. One could argue that the data here simply provide evidence that NAS scoring of inflammation provides better discrimination than SAF when compared with the same digital pathology method. The overlap and relationship should be very carefully and transparently documented.
Notwithstanding the concerns above, there are other points that should be addressed:
- The cases should not be mixed by aetiology for data presentation in tables and plots. The distinct groups – NAFLD and lobular hepatitis (not a single entity but it’s reasonable to combine AIH and viral even if METAVIR was derived for chronic HCV only) have separate scoring systems that may share a 0-3 or 0-4 scales but are completely different. Tabulating or plotting the two combined draws a false equivalence. Score 1 NAS inflammation (<2 foci of lobular inflammation per 200x field) is not comparable with score 1 METAVIR activity (itself a composite of interface and lobular scoring). Similarly, fibrosis scales for NASH-CRN and METAVIR are obviously not the same and cannot be plotted on the same graph with a shared axis.
- Even accepting that METAVIR can legitimately be used for AIH, it’s not reasonable to use it for PBC or ‘other’. These 8 cases should be removed, then NAFLD or lobular hepatitis (AIH and viral) tabulated and plotted separately.
- A diagnosis of NASH is made here based on a summation of NAS components. The NASH Clinical Research Network and the participating pathologists have explicitly stated that ‘the NAS scoring system was not intended to be used as a surrogate for a diagnostic determination of NASH versus NAFLD without NASH’[1] and that ‘NAS cannot replace a histological diagnosis’. The cases included in this study should be diagnosed by the pathologists, not the scores.
- The diagnostic categories (CPH and CAH) within lobular hepatitis that are used to determine accuracy of classification using the digital quantification methods are outdated, in part because they conflate disease activity and stage, which can and should be assessed and recorded separately, under a single heading. Instead of determining if quantification of inflammation can help determine CPH v CAH it would be better to use the METAVIR activity scores to create 2 categories – none/low v moderate/severe – and determine if digital quantification of inflammation can reach the same categorisation.
- What was the indication for the biopsies, at least within broad categories i.e. aetiology uncertain/mixed, to determine degree of scarring/discrepant Fibroscan?
- The authors have the LFTs contemporary with the biopsy, so how do they perform in classifying NAFLD v NASH for NAFLD spectrum and none/low v moderate/severe activity in lobular hepatitis? Either alone and compared with digital quantification alone or with the 2 modalities combined.
- Was the IHC done in a single batch, a small number of batches, or as-and-when? The utility of a digital pathology method can only be assessed when full details about the generation of the input images are known.
1 Brunt EM, Kleiner DE, Behling C, et al. Misuse of scoring systems. Hepatology 2011;54:369–70. doi:10.1002/hep.24347
Reviewer 2 Report
Major:
- Correlation of continuous and categorial variables (counts vs. grading) is applied (Spearman correlation “ρ”) which should be replaced by a proper statistical method of correlation. Please consider getting statistical advice on this matter.
- The authors used morphological operations to quantify cellular markers of immune/haematopoietic infiltrates (CD45 staining) and compared the performance to humans/pathologists scoring (probably sections of the same block; physical slides?) H&E staining’s. Why not using H&E staining’s to score the inflammation by applying state of the art deep learning? Please discuss limitations in the paper and point to the plethora of other image analysis techniques that already allow to answer all of these questions technically (segmentation, cell type classification on H&E).
- What is the reason for the authors to follow the proposed approach (Figure 1) instead of quantifying cellular infiltrates (positively stained cells / negatively stained cells) as measures? Please compare the metrics by analyzing your samples with open source software that is widely used by the community in order to help a given reader to reproduce the results (e.g. QuPath Bankhead, P., Loughrey, M.B., Fernández, J.A. et al. QuPath: Open source software for digital pathology image analysis. Sci Rep 7, 16878 (2017). https://doi.org/10.1038/s41598-017-17204-5; or any preferred software). Consider making your code publicly available using a GitHub repo.
- Figure 4. What is the rational to compare 0 vs 1-3 inflammation grading?
Minor:
- “Compared to NAFLD, chronic hepatitis etiologies had higher infiltration of inflammatory cells and its digital morphometric” (page 8, lines 235, 236). The authors did not quantify inflammatory cells but staining intensity areas and clusters of staining intensities within a given area.
- “First, immunostaining CD45 is non-specific of different haematopoietic cells. Other specific markers, such as CD68 and CD163 for macrophage/Kupffer cell polarization, should be addressed in further studies to better evaluate NAFLD severity “(page 9, lines 290-292). The authors should mention what cell types are stained by CD45. Why do the authors emphasize macrophages?
- Consistency: “P_Value” vs “P=..” vs. “P-value”.
- Metrics: m2; BMI=…
Reviewer 3 Report
Review of Digital Pathology Enables Automated and Quantitative Assessment of Inflammatory Activity in Patients with Chronic Liver Disease
The manuscript submitted by Marti-Aguado and co-authors presents a comparison of a traditional visual evaluation of chronic liver disease as observed in histopathology against a pipeline of image analysis. The manuscript is fairly well-written and presents interesting conclusions, that the image processing pipeline correlates well with the pathologists’ evaluation. Given the huge workload of pathologists worldwide, this is an interesting conclusion and worth communicating. The manuscript could be strengthened if the authors would address the following comments:
- Introduction is rather short. One paragraph on digital image analysis is small for a very rich area of study with many conferences (e.g. European Congress on Digital Pathology), journals and even special issues of important journals like Medical Image Analysis. Please expand, this would also help justify the pipeline that is proposed later in the manuscript
- The structure more or less follows introduction, materials and methods, results, discussion. However, the Results section starts with the MATERIALS, this description should be moved up to materials and methods
- In 2.1, line 85, please include which “Hepatology Units” are referred to.
- The image processing pipeline is simple, yet effective. It depends on certain hard values like the threshold in step c), how was this value selected? Is this transferable to other images? How was the colour standardized in step b)? Authors should be well aware of the variability of appearance of staining, and the effects of color normalization (e.g. https://www.mdpi.com/2072-6694/12/11/3337). What would be the effect of using a mask of 3 or 7 pixels in step d)? Most probably they selected this after a series of trial and errors observing the output. Authors should do a sensitivity analysis by looping over these values and comparing the results. The same should be done with the threshold.
- Fig 1 is good to illustrate the pipeline, but it is not possible to appreciate the details. They should repeat the figure of a small region of interest (top right would be good) so that the detail of the image processing algorithm can be fully appreciated.
- Fig 4 is confusing. Authors have used green and blue to associate with NAFLD and CH in previous figures and in this one they use it to correspond to inflammation and clusters. Please change. Even better would be to use different line styles (solid, dashed, …) in case the paper is printed without color.
Round 2
Reviewer 1 Report
The authors have carefully and comprehensively addressed all my comments in this new draft of the manuscript. That the methods are applied to routine IHC staining, rather than a study-specific single batch, indicates methodological robustness and that this could be applied in clinical laboratories quite easily.